# Effectiveness of Communication Interventions in Obstetrics—A Systematic Review

**DOI:** 10.3390/ijerph18052616

**Published:** 2021-03-05

**Authors:** Sonia Lippke, Christina Derksen, Franziska Maria Keller, Lukas Kötting, Martina Schmiedhofer, Annalena Welp

**Affiliations:** 1Department of Psychology and Methods, Jacobs University Bremen, 28759 Bremen, Germany; C.Derksen@jacobs-university.de (C.D.); F.Keller@jacobs-university.de (F.M.K.); L.Koetting@jacobs-university.de (L.K.); Schmiedhofer@aps-ev.de (M.S.); A.Welp@jacobs-university.de (A.W.); 2Aktionsbündnis Patientensicherheit e.V. (APS), 10179 Berlin, Germany

**Keywords:** communication, patient safety, obstetrics, midwifery, intervention, training, interprofessional, learning

## Abstract

(1) Background: Obstetric work requires good communication, which can be trained through interventions targeting healthcare providers and pregnant women/patients. This systematic review aims to aggregate the current state of research on communication interventions in obstetrics. (2) Methods: Using the PICOS scheme, we searched for studies published in peer-reviewed journals in English or German between 2000 and 2020. Out of 7018 results, 71 studies were included and evaluated in this synthesis using the Oxford Level of Evidence Scale. (3) Results: The 63 studies that included a communication component revealed a positive effect on different proximal outcomes (i.e., communication skills). Three studies revealed a beneficial effect of communication trainings on distal performance indicators (i.e., patient safety), but only to a limited extent. Most studies simultaneously examined different groups, however, those addressing healthcare providers were more common than those with students (61 vs. 12). Only nine studies targeted expectant mothers. Overall, the evidence level of studies was low (only 11 RCTs), with 24 studies with an evidence level I-II, 35 with level III, and 10 with level IV. (4) Conclusions: Communication trainings should be more frequently applied to improve communication of staff, students, and pregnant women and their partners, thereby improving patient safety.

## 1. Introduction

In obstetrics and gynecology, medical errors cause high healthcare costs and negative outcomes for women and their newborn babies [1]. Specifically, communication errors have been identified as the main cause in 72% of all perinatal deaths [2]. A key component to reducing errors and thus ensuring patient safety is promoting good patient–provider communication and effective communication between healthcare professionals [3,4]. However, communication in obstetric care needs to be improved, and this need has been reflected in public debates about obstetric violence, and in the face of emergencies [5]. It is necessary to prevent such traumatic experiences and/or employ debriefing in an emergency by means of efficient, effective and safe communication [6]. In a systematic review published in the year 2002, dissatisfaction around birth was found to be negatively related to the amount of support from caregivers, the quality of the caregiver–patient relationship, and their involvement in decision making [7].

All of these aspects can be trained by different interventions including simulation trainings. A number of single studies have identified communication as the key determinant in patient safety in obstetrics. Interventions seem to buffer effects of social inequality, lack of childbirth preparation, pain, and medical interventions [6,7]. However, there is no systematic review summarizing the single findings with a focus on the effectiveness of communication interventions. The purpose of this review is thus to provide an overview of the current state of research, to identify how communication can be improved, and to examine conclusions for further research. Since patient safety is the overarching goal in all research on communication in obstetrics, we will consider patient safety as a distal outcome along with more proximal outcomes of communication such as communication skills, ‘technical’ communication behavior, and interpersonal aspects of communication.

### 1.1. Patient Safety

Patient safety is commonly defined as the absence of preventable adverse events (PAE) or incidents [4]. Patient safety incidents are PAEs that are a consequence of healthcare processes and healthcare interventions, rather than of the patients’ condition itself. Ineffective communication between healthcare professionals (HCP) as well as between HCP and their patients can play a large role in causing such incidents and events.

Patient safety is a key performance indicator in healthcare, including obstetrics, and requires a variety of different behaviors [8,9]. Besides medical and technical skills, teamwork [10] and communication has been shown to be an integral component of safe patient care [4]. For this reason, teaching communication skills has been implemented in medical curricula, and communication trainings are a regular component in continued education [9]. Overall, there is evidence that communication trainings might reduce PAE [11,12]. However, the obstetric setting, and particularly the birthing process, requires more detailed attention for several reasons.

Usually, the role of the expectant mother is different from the role of a patient who has an illness that needs to be cured. The hospital situation and the participation of the expectant mother and her partner are connected to the anticipation of parenthood. Many expectant mothers expect that their anticipation will be met by HCP. While nervousness and pain are, to a certain degree, considered a part of this process, the overall idea of the hospital stay is of excitement and joy. However, although pregnancy and childbirth are not usually pathological processes requiring medical interventions, potential (severe) complications require effective interprofessional (medical) interventions and collaboration [13]. Maintaining good interpersonal patient–provider communication despite the stressful situation is crucial, e.g., to reduce the risk of psychological trauma by keeping everyone informed and thereby enabling the pregnant woman/mother to express concerns [1].

For these reasons, communication trainings from other medical fields might not be transferable to this setting [14]. Tools and techniques need to be adapted to the specific context to ease the transfer of learned skills and pre-existing knowledge into practice. Consequently, communication trainings for HCP have been adapted to the obstetric setting in recent years. Additionally, medical and nursing students are often required to practice communication skills in obstetric settings during their education. However, despite the integral role of the expectant mother in this setting, efforts to actively engage her in communication processes have been scarce [15].

### 1.2. Communication in Obstetrics

Multiple models of safe healthcare communication exist that might be transferred to obstetric care. Accurate, clear communication is central to all of them. For instance, communication is one of four central aspects in the TeamSTEPPS framework of teamwork in healthcare, which has been validated in many clinical settings including obstetrics [16]. The communication dimension of this framework utilizes different tools to facilitate safe communication between healthcare providers, particularly to ensure the recipient has understood the sender’s information correctly (e.g., closed-loop communication), to convey critical information to a larger group of people efficiently (e.g., callout), and to request help in emergencies in which information needs to be conveyed quickly (e.g., the Situation Background Assessment Recommendation, SBAR, technique or checklists) [17].

Focusing on transmitting factual information is necessary for effective communication but is sometimes deficient, especially in obstetrics. Interpersonal and relational communication has been identified as one of four dimensions in midwifery models of care [18]. These models consider that—unlike most other reasons to visit a hospital—birth is a natural process in most cases and a joyful moment for the parents, but it can also cause uncertainty or fear. Thus, consideration of interpersonal or relational aspects of communication with the expectant mother—such as being respectful and taking her emotions into account—is essential to convey important information to the expectant parents. Moreover, focusing on interpersonal communication with other team members is likely to improve team functioning and increase team members’ motivation to engage in communication [19].

When evaluating communication competencies in healthcare, teamwork also needs to be examined: in a previous meta-analysis, teamwork was revealed to be positively related to performance [10]. While both are important training outcomes and determinants for patient safety, communication is crucial for team functioning. Supposed technical and emotional communication competencies are exhibited appropriately in the team setting. In that case, the processes of encoding, decoding, and transactional sense-making are performed adequately, thus increasing the likelihood of safe communication [4]. As a consequence, the risk of PAEs may be reduced. Therefore, in this study we focus on communication as a part of teamwork but consider teamwork training only if it includes a specific communication aspect (including digital interventions and simulation training) [20]. As no systematic review or meta-analyses could be explicitly found on this aspect, but only on simulation training in obstetrics [6], teamwork in general [10,19], communication training in nursing care [14] or midwifery [21], we fill this gap by investigating all disciplines working in obstetrics (not only midwives or nurses) and including all communication training approaches (not only simulation or teamwork).

### 1.3. Research Aims

Based on a rather broad definition of communication, the goal of this review was to summarize and evaluate the current state of research on communication interventions in obstetrics. Communication interventions include trainings for HCP, interdisciplinary teams, expectant mothers, and their accompanying persons. Thus, our general research question was formulated as follows: Do interventions including a communication component have an effect on proximal (i.e., communication skills) or distal (i.e., patient safety) performance indicators in obstetrics? According to the PICOS scheme (Population, Intervention, Comparison, Outcomes, and Study design [22]), we aim to evaluate the following research questions in detail:Participants: (1a) Are communication trainings more frequent during education or on the job? (1b) Are expectant mothers (and their accompanying persons) also targeted?Interventions: (2a) Are the interventions grounded in theory or organizational initiatives? (2b) Are communication trainings typically standalone interventions, or are they part of larger training programs?Comparisons: (3a) Which conclusions regarding the strength of effects can be drawn from the comparisons analyzed in the original studies? (3b) Are single or interactive effects investigated?Outcomes: (4a) What types of outcomes are examined? (4b) Are distal outcomes such as patient safety considered?Study design: Based on the study design, how robust are the results?

While the aim is to synthesize the evidence from the original studies, different factors relating to communication or target groups will not be separated. The included target groups, outcomes, study designs and characteristics can vary widely across publications and will therefore only be aggregated by means of a systematic review and not a meta-analysis.

## 2. Materials and Methods

### 2.1. Definition of Central Concepts

Communication was defined as broadly as possible in order to avoid missing important publications in the field. Therefore, all interventions targeting the exchange of information were included if the manuscripts matched other inclusion criteria (see below). Publications were included when either the intervention was focusing only on communication, or when a teamwork intervention contained an explicit communication component. We regarded both relational (such as acknowledging the patients’ perspective) and technical (such as completeness of clinical information) communication interventions. As communication in healthcare is transmitted via various channels, we considered oral, written, analog, and digital forms of communication.

The term intervention was defined widely, too. We included all structured actions intended to improve communication in obstetric settings. Thus, interventions in the strict sense (team or communication training for a scientific or practical purpose) and also large-scale quality improvement initiatives, examinations in medical/nursing education, or implementation of technology all fell under our definition of intervention if they focused on the obstetric field.

### 2.2. Search Strategy

We searched six databases (web of science, psycinfo, cinahl, medline/pubmed, eric, cochrane systematic reviews) to identify relevant literature (Figure 1). The cochrane systematic reviews database was searched to identify previous relevant systematic reviews and scan them for additional relevant studies. The basic search term was ((communication OR team) AND (training OR intervention OR skills)) AND obstetric*. The full search strategy can be found in Appendix A.

We searched for titles, abstracts, keywords, and journal titles since publications in obstetric journals might not mention the word ‘obstetric’ in titles, abstracts, or keywords, which could have led to the exclusion of relevant publications. In addition, we also searched for the appropriate MeSH (Medical Subject Headings) and thesaurus terms. In this first stage, we included studies published in peer-reviewed journals in English or German between January 2000 and December 2019 extensively and performed an update until the end of November 2020. Studies in the German language were included as the study was funded by a research funding and decision making body in Germany interested in local evidence, too. No further languages were included as no sufficient professional language proficiencies existed other than English and German in the research team. We also hand-searched reference lists of identified systematic reviews in our initial search to identify additional eligible studies that were included as separate references in the review at hand. Studies from the last 20 years were included to get a large overview on developments in the trainings and evidence from the last two decades. This approach should level out outliers related to timing effects such as policy changes in specific countries, economic crises of specific regions or global challenges such as the COVID-19 pandemic.

### 2.3. Screening and Selection Procedure

After excluding duplicates, non-peer-reviewed publications, and publications in languages other than German or English, two raters (A.W. and either N.H., N.L., N.S., or S.L.) screened all references independently. At the title and abstract screening stage, we included empirical (i.e., qualitative and quantitative) studies referring to communication or team interventions in an obstetric setting. ‘Obstetric settings’ include studies centered around pregnant women and women trying to get pregnant (and their partners), the birthing, and the women’s postpartum stage. Simulation trainings of the mentioned situations were included. Studies that focused on newborns were included if the situation described in the study was part of the birthing process (i.e., resuscitation of a newborn immediately after birth). Otherwise, these studies were considered as pediatric settings and therefore excluded.

In terms of study participants, pregnant women, women trying to become pregnant and their partners, obstetric healthcare professionals (i.e., midwives), healthcare professionals working in an obstetric setting (such as anesthetists), and students working in an obstetric setting were included. We decided to include educational settings with students to account for the fact that nontechnical skills (NTS) training has been integrated into medical and nursing curricula in recent years. In doubtful cases, i.e., if relevant information was missing, studies were included for full-text screening.

At the full-text screening stage, we included studies that (1) explicitly implemented a communication intervention and measured change in relevant outcome variables. These relevant outcome variables could focus on communication or other constructs, such as clinical parameters thought to be changed by communication training (e.g., EmONC-simulation curriculum by Afulani et al., 2020 [23] or VitalTalk by Chung et al., 2020 [24]). We also included studies that (2) described communication as part of a team training intervention and measured change in a communication-related outcome variable (the effect cannot be attributed to communication only).

Outcome variables could be measured via a pre–post comparison (i.e., follow-up over time) or subjectively reported improvements of relevant variables. Thus, all levels of evidence from randomized control trials (RCTs) to qualitative interviews were included. Even though the evidence for an intervention-induced change in descriptive or qualitative studies cannot be compared to RCTs, these studies nevertheless provide a more in-depth understanding of current research questions and gaps. Thus, also investigations with just one measurement point (“one-shot” initiatives) were included. Studies that did not provide descriptions detailed enough to judge whether a study belonged to case (1) or case (2) were excluded (Figure 1).

At the title and abstract screening stage, all articles selected by either rater were included. At the full-text screening stage, disagreements were solved by reaching a consensus through discussion. An overview of the screening and selection procedure can be found in Figure 1.

### 2.4. Quality Rating

The quality rating consisted of two steps. In the first step, we rated studies based on an adapted version of the levels of evidence defined by the Oxford Centre for Evidence (OLE) to determine the robustness of a study’s findings [25]. However, this instrument provides a global overview and is generally not suitable to rate qualitative studies, which we included to gain a more in-depth understanding of healthcare professionals’ perception and acceptance of interventions. Thus, in order to systematically evaluate all included studies, we combined and adapted previous scales [25], which assessed dimensions such as transparency of reporting or the appropriate use of the methodology chosen. Each study was rated by two independent raters (A.W. and N.H., N.S., or S.L.). Not all items were applicable to all studies. Disagreements were resolved through discussion.

### 2.5. Data Extraction

We extracted information on the study setting and population, variables of interest to this review and their measurement, type of intervention and analysis, main results, and whether the communication was the focus or one part of the intervention from the reviewed studies.

## 3. Results

After completing the screening procedure, 71 studies were included (see Table 1 for details).

### 3.1. Study Characteristics According to the PICOS Scheme

Investigating research question 1a, the majority of interventions addressed study participants who were professionals (HCP) with 65 publications (91.5%). More specifically, 35 studies aimed to improve communication in interdisciplinary teams, 12 in residents, nine in midwives, four in nurses, three in anesthetists, and two in other healthcare providers. Twelve communication interventions targeted students (16.9%), with five addressing medical students, four nursing students, two midwifery students, and one training a general student group. Nine studies included mothers, pregnant women, or patients (research question 1b).

Testing research question 2a, whether the interventions were grounded in theory or organizational initiatives, we found the authors reported theory-based approaches in all studies (using implicit or explicit theories concerning simulation, communication, shared decision making, skills training, and error disclosure). This approach was more frequent than organizational initiatives (patient safety culture, safety interventions/training, and organizational targets) with just six studies.

Answering research question 2b, 40 studies (56.3%) solely focused on communication interventions, whereas in 31 studies (43.6%), communication was part of a team training or other type of intervention such as addressing organizational aspects. Addressing research question 3 regarding the comparison, the majority of studies (40 publications, 56.3%) employed a pre–post design. In total, 14 studies (19.7%) used a retrospective post-intervention evaluation. Twelve studies (16.9%) implemented an RCT design. One study employed a control group but failed at randomization [83]. One publication was a systematic review that aggregated studies from simulation trainings [6].

Regarding research question 3a, 63 studies revealed a positive effect on different proximal outcomes (i.e., communication skills). Three studies evaluating the effect of communication trainings on distal performance indicators (i.e., patient safety) proved to be beneficial to some extent. Relating to research question 3b, all included studies evaluated single effects, and additionally, only four studies’ interactive effects were evaluated explicitly [45,69,74,81]. Detailed results are reported in Table 1. We identified 37 publications (52.1%) that used questionnaire data. Twenty-three studies employed observations as the primary outcome measure (32.4%). Eighteen studies (25.3%) used qualitative approaches to gain an in-depth understanding of participants’ experiences with the intervention and perceived learning. Four studies (5.6%) measured outcomes with recorded reviews in pre–post designs or RCTs [20,28,43,53].

Regarding the measured outcomes (research question 4a), fifty-eight studies (81.7%) investigated ‘technical’ communication, i.e., structured or standardized communication during medical procedures or processes, such as information seeking or decision making. Additionally, four studies focused on interpersonal aspects of communication with patients or colleagues explicitly, such as establishing a relationship. A more frequent outcome measure included 16 clinical aspects of communication including, for instance, neonatal resuscitation, emergency simulation or complete and accurate transmission of medical information.

The 37 studies employing questionnaires explored preferences, self-rated skills, and assertiveness, with no predominant survey instrument, and focused mainly on communication and adherence to standardized procedures. Observational studies examined ‘technical’ communication as the primary outcome, adapting one of the validated nontechnical skills (NTS) observational tools to the obstetric setting. The four studies using retrospective record reviews evaluated the completeness of written communication/information (e.g., location of delivery), information transmission to better prepare women for childbirth, and compliance with standardized handover protocols. Qualitative studies focused mainly on ‘soft’ communication aspects such as empathy and nonverbal communication, reducing hierarchies as well as encouraging open discussion of cases. Only three studies investigated adverse effects or errors by capturing the number and reporting of adverse events in organizational data [11,12] or by using the Adverse Outcome Index (AOI) [72]. Summarizing this regarding research question 4b, distal outcomes such as patient safety were only rarely considered.

### 3.2. Intervention Effects

Most studies revealed some positive effects of an intervention, with a majority revealing moderate effect sizes (40 publications; 56.3%). Only 14 investigations (19.7%) revealed effects as hypothesized. Nine studies (12.7%) presented positive effects but used only qualitative approaches so that an effect size could not be reported. Only 10 studies did not find any positive effect (14.1%). In the Appendix A, we describe representative study designs grouped into highly effective interventions (Appendix B), those with moderate effects (Appendix C), qualitative research design (Appendix D), and studies not finding any effects (Appendix E). These effects appeared on an explorative level unrelated to the time of publication of the study.

### 3.3. Study Quality According to the Oxford Level of Evidence

In order to give an overview of how robust the results are (PICOS scheme) and to examine research question 5, the Oxford Level of Evidence scale was applied. Overall, only one study was rated as level Ia (1.4%). Ten studies were categorized as Ib (14.1%), one as IIa (1.4%), and 13 as IIb (18.3%). Twelve studies rated as IIIa (16.9%), and the majority (23 publications) could be classified as IIIb (32.4%). Additionally, 10 were rated as IV (14.1%).

Study designs with a higher level of evidence (Ia, Ib, IIa, and IIb) all demonstrated positive effects of their communication interventions. Study designs with a low evidence level and thus lower quality (IV) were more likely to not demonstrate any effects (8 out of 10 studies failed to reveal positive effects, Figure 2). Only one out of all the studies with a medium level of evidence (IIIa + IIIb) was unable to show a positive effect. Notably, not only more but also stronger effects were found in the Ia and Ib evidence level studies than in the IIa and IIb evidence level studies when examining effect sizes (Figure 2). Summarizing the findings on research question 5, the study designs revealed partially robust results.

Methodological limitations that affected the robustness of evidence occurred in all studies that did not feature a control group, randomization, or assessment of control variables. Small sample sizes and large dropout rates prevented the calculation of statistics in some studies. Other limitations included the lack of pre–post comparisons (i.e., missing baseline) or follow-ups. In one study, pre- and post-samples were different: one group of patients was assessed before HCP were trained and a different group after the training, so effects could have also been caused by differences in groups.

Small sample sizes might be a reason for the lack of reporting regarding interaction effects in training evaluation. Only six publications reported interaction effects (8.5%), with only one study reporting a long-term follow-up. In this study, OB/GYN residents were assessed in a prospective cohort pilot study. After 3 months, it was evident that intervention effects were maintained at the post-intervention level over 3 months, but a further increase could not be attained [24].

Moreover, there was no clear description of how communication was trained or operationalized in the intervention in many original studies, which seemed especially problematic in qualitative studies: Most qualitative interviews, observations, or objective data lacked accurate and comprehensive reporting. Studies, however, using subjective data, would also require (more) validation or additional evaluation of organizational or objective data. A possible bias in subjective data relates to social desirability, which was rarely controlled. Moreover, convenience sampling methods should be overcome by more appropriate sampling methods and striving for recruiting representative samples.

## 4. Discussion

A systematic review was performed to synthesize the evidence on training expectant mothers before and during the birth process, and interprofessional communication skills [21,88]. As various aspects of the field of obstetrics were of interest (from shared-decision making to emergency training), our searching strategy was based on a broad definition of communication and combined different participant groups, interpersonal interactions, and communication tools.

With this systematic review, we aimed to aggregate the current state of research on communication interventions in obstetrics up until the end of the year 2020. This was done by looking at different target groups as this had not been done before. Previous systematic reviews within obstetrics generally focused on the training effects on teamwork and team performance [19], while in obstetrics, only hybrid simulations but no other forms of communication training were addressed [6]. Although there are many studies including systematic reviews and meta-analyses, most of them simply aggregated interrelations of communication [1,7,10]. The previous reviews aggregating communication training programs match our findings but they were performed only generally in midwifery [21], nursing care [14], or student learning [25] but without an interdisciplinary and intersectoral approach as with our study. Thus, our systematic review expands the previous state of science and will be synthesized in more detail in the following.

### 4.1. Overall Results

The overarching research question of this review was whether interventions including a communication component have an effect on (a) proximal (i.e., communication skills) or (b) distal (i.e., patient safety) performance indicators in obstetrics. Having provided an overview of interventions with high, moderate, and no effects, we can conclude that interventions including a communication component are effective for proximal outcomes.

Although communication was operationalized differently between studies, nearly 20% of all studies found effects as previously hypothesized regarding ‘technical’ or ‘relational’ communication skills. Another 56% found moderate effects indicating that the communication training had a positive impact on the trained study population, although either different or smaller than had been hypothesized. In nine studies, only qualitative indicators were given such that effect sizes could not be computed/inferred. However, these studies can provide a more in-depth understanding of the mechanism of how training improves skills or other communication outcomes. This may also be valuable in understanding the effects of training, for instance, by demonstrating that many HCP were not aware of the importance of communication for patient safety prior to the communication training [40].

So far, conclusions can only be drawn with great caution: only three studies targeted patient safety outcomes, such as adverse events, achieving mixed results. One study provided evidence that the reporting and occurrence of PAEs could be reduced; whereas another study did not find positive effects. Therefore, it is clear that more studies aiming at reducing PAEs and thereby improving patient safety are needed. Only then can possible mechanisms be identified to inform new and promising approaches. However, more distal patient safety outcomes would plausibly be more objective but at the same time also be more expensive. This should be taken into account when planning future research.

Another important consideration limiting the conclusion that communication training is generally effective is that in nearly 44% of publications, communication was an integral part of a broader teamwork training approach. It is correspondingly unclear whether the communication intervention alone improved communication skills or whether a team training framework is needed to achieve improvements in communication.

However, a majority of studies (56%) strongly focused on communication as a standalone intervention, thus indicating that the communication component is crucial. Nevertheless, more research in terms of dismantling studies is needed. The same holds true regarding the relationship between the time of publication of the study and the effectiveness of the training. It would be worthwhile determining whether trainings improve over time due to more aggregated evidence feeding into improved training developments. Clearly, evidence should be used to inform future trainings and their evaluations.

### 4.2. PICOS Research Questions

It must be borne in mind that study designs and characteristics strongly differed between publications. Overall, our review presents studies with heterogeneous approaches regarding study participants, intervention methods, study design, outcome measures, and operationalization as well as study quality. Despite the common knowledge of the importance of communication for patient safety in obstetrics, it has yet to be systematically trained in mothers.

Additionally, there are almost no studies with effects on patient safety measured directly. However, the majority evaluates proximal outcomes regarding communication skills. In the following, we provide an overview of how different study characteristics seem to impact the effectiveness of communication training according to the PICOS scheme.

#### 4.2.1. Study Population

Concerning the frequency of communication trainings, professionals who had finished their education were targeted more frequently. According to our results, students were explicitly addressed in only a few studies. As communication is an essential aspect of patient safety, it should be trained at an early stage of education [54,89]. Furthermore, to reach sustainable improvement, it is recommended to be repeatedly provided on a regular schedule for all stages of the career [90]. However, most studies implemented trainings that was not repeated or advanced.

Considering that expectant mothers were rarely addressed, and that it is of crucial importance to gain a mutual understanding of their needs [4], we suggest including them more often. Expectant mothers and their partners should be targeted to improve communication skills and assertiveness within the birth process. A possible challenge is that expectant mothers usually only have limited contact with the facility in which they plan to deliver their babies, thus making it difficult to address them with training prior to childbirth. This challenge is even more pronounced due to the current COVID-19 pandemic. Possible solutions include the adaptation of digital information or interventions. Creating digital possibilities (e.g., online training or apps) might be beneficial regarding the patient perspective [20], caregiver–patient relationship and satisfaction around birth [7].

Furthermore, we encourage the promotion of study population characteristics related to their background and culture, as the health service context should be considered. This is essential because obstetric teams in developing countries face different challenges relative to developed countries and high-income regions [91,92]. In high-income countries and areas, barriers towards patient safety and communication are very different than in countries in which the absence of highly necessary equipment cannot be taken for granted [93].

#### 4.2.2. Intervention Characteristics

Our findings show that many interventions were part of a more comprehensive team training, e.g., simulation program, in which improved communication skills were one out of many objectives. These results indicate that simple communication interventions should be integrated into broader team trainings, especially into those featuring a simulation of crucial situations. However, the current COVID-19 pandemic poses a significant barrier to the implementation of well-planned, repeated, and targeted training programs since infection prevention measures have to be met at all times. Particularly in the near future, digital interventions (telehealth, eHealth, mHealth) open new avenues [4,20,64].

Potentials of digital modes become especially important in times of work concentration due to increasing efficiency, fewer experts on the labor market, and multitasking as a societal trend [94], which are long-term challenges in healthcare, including obstetrics. There are multiple advantages to digital training, such as adaptability to the user’s needs [4], just-in-time-interventions [95], and a high number of potential users who are motivated to participate due to technological interests instead of the content, making it easier to reach unmotivated individuals [4,20,64,83]. Another benefit of digital interventions is that they might compensate physical distance due to the pandemic or living remotely with limited access to training options.

Concerning intervention techniques, it must be noted that few of them were described in detail. Since the availability of intervention protocols is an important aspect of Open Science, aiming to increase the transparency and thus reproducibility of interventions and evaluations [96], future research should aim to provide more accurate and clear descriptions of their design and interventions.

#### 4.2.3. Comparisons and Analyses

On the positive side, we can conclude that most of the pre–post comparisons indicate that trainings were effective. However, only 12 studies employed an RCT design, while the majority of intervention studies used designs with lower evidence levels. In addition, many lack a description of respective significance of results. Furthermore, only a few interaction effects were reported.

Training outcomes were oftentimes only measured in a post-treatment time point of measurement, with only one study providing more long-term evidence [24]. These results indicate a need for further well-planned, high-quality interventions with a clear description of training topics, methods, and corresponding outcomes. What appeared to be promising, however, was that study quality was related to the occurrence of positive effects overall in publications. Out of the study designs with a higher level of evidence (Ia to IIb), zero reported no or negative effects, while studies of lower quality were more likely to demonstrate no positive effects (8 out of 10 in evidence level IV).

#### 4.2.4. Outcomes

As mentioned above, the outcomes are mainly examined as proximal outcomes of the interventions (i.e., improved communication skills) while only a few studies tested for distal indicators such as patient safety or culture of error reporting [11,12,72]. However, the variety of measured outcomes, including correct hand-over information, the support of assertiveness, openness, and interpersonal communication, mirrors the vast potential for optimization.

As different aspects of everyday clinical practice are addressed, it is difficult to draw clear conclusions regarding the improvement of communication and overall patient safety. However, the definition of communication applied in this research already reflects the broadness of communication skills and techniques. Although the differentiation between ‘technical’ and ‘relational’ skills offers a first leverage point to frame communication, a comprehensive model of necessary skills in obstetric care has, to our knowledge, yet to be established. Defining a set of skills as well as a framework modeling the mechanisms could help to identify shortcomings and standardize communication training so that the most effective interventions could be further developed.

The same applies to the assessment and evaluation of PAE. Up to date, there is no general understanding of which events can be classified as a PAE in the obstetric setting, which makes it more difficult to identify and target determining factors. Therefore, an agreed-upon approach towards patient safety is needed, especially in obstetrics, gynecology, and women’s health [91].

#### 4.2.5. Robustness of Study Results

As described in Section 3.3, about half of the studies (35 of 71) included were classified as level III based on the Oxford Level of Evidence (OLE). While studies with a higher quality also revealed better results, we can conclude that the validity of effects drawn from most comparisons analyzed in this review is mixed due to the different designs, which also differed in quality. Furthermore, only the main effects but very few interaction or mediating or moderating effects were investigated so far. In this regard, we can conclude that the results of the aggregated studies appear robust in terms of general positive effects, but we expected to find stronger and higher quality in original studies. It appears that studies with a higher quality also have a higher chance to actually detect positive effects. Thus, the clear recommendation is to conduct well-planned studies and interventions.

All qualitative studies were assigned to level IV of the OLE as they mainly focused on additional aspects relating to communication such as empathy and nonverbal communication, reducing hierarchy, and encouraging open discussions of cases. Although most qualitative publications reported some positive change, it should be taken into account that participants usually rate interventions favorably and perceive them as useful in terms of learning, working in a team, and communicating efficiently.

Only a few studies investigated adverse effects or treatment errors by capturing the number and reporting in organizational data. Qualitative studies specifically lack comprehensive reporting, which is required for classification in OLE. This could be improved in the future. Thereby, qualitative designs can contribute substantially as they can be understood as the best suitable approach to gain a more in-depth understanding of the results of quantitative surveys or to exploit a research topic through subjective perspectives and cocreation [3,13]. Therefore, qualitative results alone are not meant to be generalized in terms of quantification, but they contribute to a better understanding of behavior patterns and underlying mechanisms such as experiences, emotions, and cognition [97]. A systematic, qualitative review (e.g., qualitative meta-synthesis) would be beneficial to summarize results in the future. However, more high-quality original studies are needed for that, and for aggregating the quantitative results into a meta-analysis as well.

### 4.3. Limitations

This study presents an overview of the current state of intervention studies on communication interventions in obstetrics published in English or German language. Therefore, it is likely that relevant results published in other languages were not covered. However, English is the natural language of research in which most results are published, which is why the overview should reflect most sources available to researchers in the obstetric field.

Another common limitation to nearly all systematic reviews and meta-analyses is the publication bias: since we only searched for published articles in scientific search engines, we were not able to include studies that were never published in a peer-reviewed journal. However, it is more likely for interventions resulting in improvements to be published, so our review might show a more positive and promising pattern than is actually existent.

Furthermore, as we included a broad range of studies covering several topics on communication improvement in obstetrics, we may have overlooked significant aspects detected in single studies. An example is the background and setting of the study protocols as well as the study population. Accordingly, it is very likely that obstetric teams in developing countries face different challenges and, therefore, must conduct research as well as training under different circumstances [91,92]. However, even in high-income countries, birth settings vary: a delivery in a high-level perinatal center requires a different approach from obstetric teams and the expectant mother than delivery in a birthing center or even at home. In future studies, cultural background and international or cultural diversity of the patient–provider team as well as among the healthcare professionals should be taken into account. Language difficulties may arise but not be overcome by simple digital devices such as translators. This needs to be taken into account, especially in times of high migration rates and recruitment of staff abroad.

Finally, it was especially not possible to conduct an in-depth analysis of qualitative research in the field, so that the focus of this review was to understand the effects and their moderators. Nevertheless, qualitative studies can add a deeper understanding of possible mechanisms and should thus be combined with quantiative research to a greater extent [98]. Likewise, with the quantitative studies, analyzing and aggregating effect sizes with a meta-analytical procedure should be applied, which was not possible in this study due to too few appropriate original studies.

## 5. Conclusions

This systematic review provides an overview of important aspects of obstetric communication training and thus suggestions for future research: a large majority of the intervention studies indicate a positive effect on proximal outcomes in obstetrics, such as communication skills or behavior. However, communication training seems to be more effective in combination with team training featuring a simulation of crucial, time-critical obstetric situations. The evidence regarding patient safety and thus, the reduction of PAE due to communication training, is low because not many studies used clinical parameters as outcome measures.

Experienced staff were trained in more studies than students, and few interventions included expecting mothers. This emphasizes the need for broader, ongoing training programs targeting all staff members in all levels of education. Communication training should be provided to students, educated staff on all professional levels, and expectant mothers and their partners to improve communication and thereby improve patient safety.

With regard to future research, we strongly recommend more high-quality research. A lack of evidence still exists with regard to dismantling studies and also digital interventions [20]. This gap should be addressed by applying standards of Open Science so that interventions and designs can be reconstructed and replicated in the future, and to also test whether these effects can be replicated in other settings relating to health promotion and prevention, clinical care and medical rehabilitation.

## Figures and Tables

**Figure 1 ijerph-18-02616-f001:**
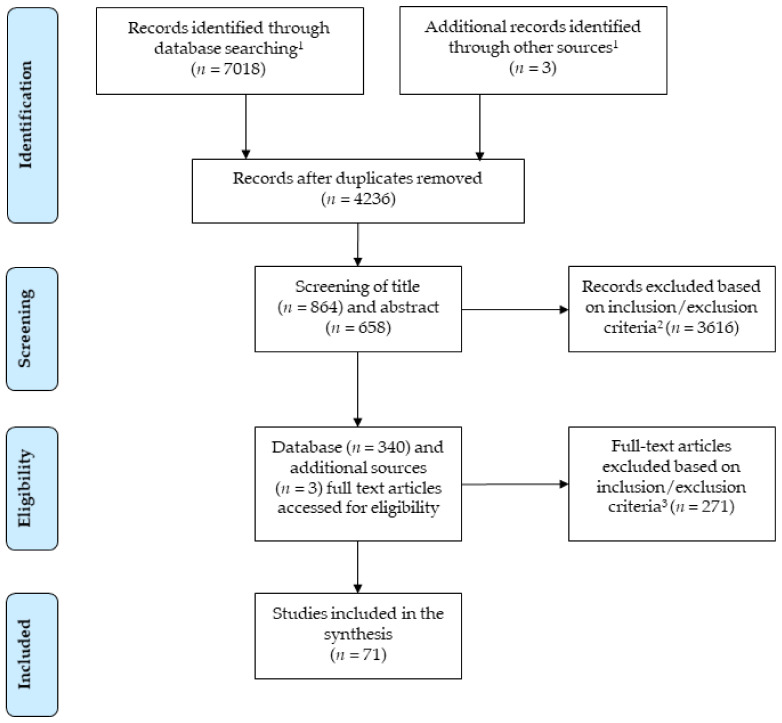
Systematic search method and inclusion/exclusion criteria. Note: ^1^
*Inclusion criteria*: Publications in English or German; published between 2000 and 2020; human subjects; peer-reviewed journal. ^2^
*Inclusion criteria*: Clinical or educational obstetrics setting; communication as an intervention or outcome variable; empirical data. ^3^
*Inclusion criteria*: Communication intervention or an intervention that includes communication or that measures the change in communication; *exclusion criteria*: No empirical study (e.g., editorial, opinion, abstract); incomplete or unclear reporting of key information (e.g., construct definition, measurement methods, analyses, statistical indices); duplicate publication; wrong setting or no obstetrics-specific data; no intervention; no change measured.

**Figure 2 ijerph-18-02616-f002:**
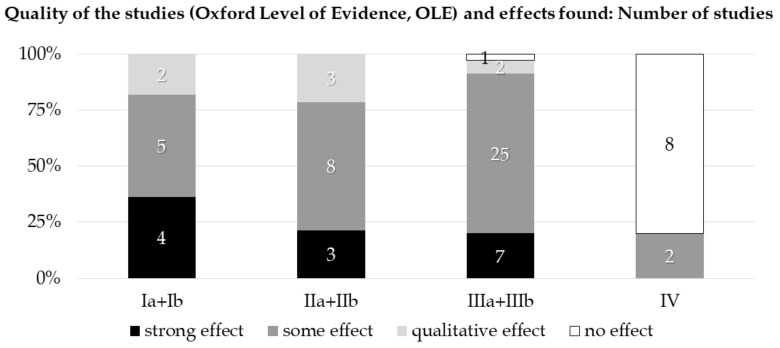
The number of studies rated by their level of evidence as an indicator for the robustness of a study’s findings.

**Table 1 ijerph-18-02616-t001:** Overview of all intervention studies on communication in obstetrics included in the current systematic review.

	Author	Country	Focus ^1^	Study Population ^2^	Methodology	Intervention ^3^	Description of Communication ^4^	Outcome Measure	OLE ^5^	Effect: Improvements	Effects: Reductions	Aggregated ^6^
1.	Afulani et al., 2019 [15]	GH	Other	M, HCP	Pre–post	TT emergency simulation	HCP’s Comm quality rated by M	Quant. survey	IIIb	Comm		++
2.	Afulani et al., 2020 [23]	GH	*Comm*	N, M, R	Pre–post	RMC-focused simulation	Emphasizing respect for feelings, and preferences	Mixed method	IIIb	Knowledge, self-efficacy		++
3.	Ahmed et al., 2019 [26]	PK	Other	R	RCT	TT (NTS)	Cesarean section, Comm	Observer rating	Ib	Comm		++
4.	Alder et al., 2007 [27]	CH	*Comm*	R, M	RCT	Comm, simulation	MAAS-R, Patient satisfaction with comm	Observer, survey	Ib	M satisfaction		+
5.	Amoakoh et al., 2019 [28]	GH	*Comm*	M, HCP	Pre–post	Comm	Completeness of written Comm	Record review	IIb	Comm		+
6.	Baijens et al., 2018 [29]	NL	*Comm*	M	Pre–post	Shared decision making	Preferences in 1. decision making and 2. information	Mixed-Method	IIIa	-	-	-
7.	Bambini et al., 2009 [30]	US	Other	S	Pre–post	Simulation of obstetric	Quality of Comm	Qualitative survey	IV	Self-reported skills		+
8.	Bashour, 2013 [31]	SY	*Comm*	M, R	RCT	Comm	Comm, Satisfaction	Observer, survey	Ib	-	-	-
9.	Black, 2018 [32]	GB	Other	T	Retrospective	T training, simulation	Subjective Comm skills	Quant. survey	IIIb	Comm		+
10.	Bloomfield et al., 2020 [33]	CA	*Comm*	R, N, T	Pre–post 14 m	Simulation	Comm skills, knowledge	Mixed method	IIIb	Comm, knowledge		+
11.	Bonnema et al., 2009 [34]	US	*Comm*	R	Retrospective	Comm	Comm regarding error disclosure	Quant. survey	IIIa	Comm		+
12.	Cavicchiolo et al., 2019 [35]	MZ	Other	M	Pre–post	Clinical skills (neonatal resuscitation)	Comm per ANTS	Observer rating	IIb	-	-	-
13.	Chung et al., 2020 [24]	US	*Comm*	R	Pre–post 3 m	VitalTalk simulation	On-the-spot encouragement, suggestions Comm	Quant. survey	IIIb	Comm		+
14.	Crofts et al., 2008 [36]	GB	Other	T, patient actors	RCT	T training simulation	Comm (more information not available)	Quant. survey	Ib	Comm		++
15.	Dadiz et al., 2013 [37]	US	*Comm*	HCP	Pre–post 3 years	T training simulation	Comm, information exchange	Quantitative, observe.	IIb	Comm, info exchange		+
16.	Deane et al., 2015 [38]	IE	Other	S, M	Retrospective	Clinical skills training	Comm	Mixed method	IIIb	Comm		+
17.	Fransen et al., 2012 [39]	NL	Other	T	RCT	T training, simulation	Comm incl. SBAR, closed-loop, CTS	Observer rating	Ib	Comm		++
18.	Franzon et al., 2019 [20]	BR	*Comm*	M	RCT	E-health intervention	Information transmission	Quantitative, Record	Ib	Feeling prepared, knowledge		+
19.	Freeth et al., 2009 [40]	GB	Other	T	Retrospective	T training, simulation	Effective Comm, information sharing	Qualitative interviews	IV	Awareness		+
20.	Gardner, 2008 [41]	US	Other	T	Retrospective	T training, simulation	Comm (closed-loop, speaking-up, error dis)	Quant. survey	IIIa	Comm		+
21.	Guimond et al., 2019 [17]	US	*Comm*	S	Pre–post	T training, simulation	SBAR	Observer rating	IIb	SBAR performance		++
22.	Haller et al., 2008 [42]	CH	Other	T	Pre–post	T training (CRM)	Comm (speaking-up, asking, closed-loop)	Quant. survey	IIIa	Comm		+
23.	Hughes et al., 2017 [43]	US	*Comm*	T	Pre–post	Three interventions (phone, two digital texting)	Communicating critical delivery information (e.g., re delivery room)	Record review	IIIb	Comm		+
24.	Hughes et al., 2014 [44]	GB	Other	S	Retrospective	T training, simu. PROMPT	Not described	Qualitative interviews	IV	Comm		+
25.	Hullfish et al., 2014 [45]	US	*Comm*	T	Pre–post	Timeout checklist	Speaking-up, voice	Quant. survey	IIIb	Own opinions respected by others		+
26.	Kahwati et al., 2019 [46]	US	Other	T	Pre–post	T work and Comm	Comm; modified adverse outcomes index	Mixed-meth	III	Comm		+
27.	Karkowsky et al., 2016 [47]	US	*Comm*	S	RCT	Comm, simulation	Verbal, nonverbal and patient-centered Comm	Quant. survey, Observat.	Ib	Comm		+
28.	Kim et al., 2012 [48]	KR	*Comm*	S	Pre–post	Comm, simulation	Comm skill	Quant. survey	IIIa	Comm		++
29.	Kirschbaum et al., 2012 [49]	US	*Comm*	R	Pre–post	Comm, simulation	Comm culture as per PRIOR	Quant. survey	IIIa		Indepen dence, dominance	+
30.	Kumar et al., 2016 [50]	AU	Other	M	Retrospective	Clinical simulation	Increase in learning related to Comm	Qualitative survey	IV	Comm		+
31.	Kumar et al., 2019 [51]	IN	Other	M	Retrospective	Clinical simulation	Increase in learning related to Comm	Qualitative survey	IV	Comm		+
32.	Lavelle et al., 2018 [52]	GB	Other	T	Retrospective	T training, simulation	Described implicitly (NTS)	Qualitative survey	IV	Knowledge, awareness		+
33.	Le Lous et al., 2020 [6]	FR	Other	M, S, R	Systematic review	Simulation training	Good/efficient Comm incl. providing sufficient items	Self-report, objective	Ia	Non-technical skills		+
34.	Lean et al., 2017 [53]	GB	*Comm*	T	Pre–post	Comm training	Compliance with standardized handover	Record review	IIb	Compliance		++
35.	Lee et al., 2018 [54]	US	*Comm*	T	Pre–post	Comm intervention	Closed-loop Comm, adherence stand. handover	Organizational data	IIIb	Compliance	Durat. admission	++
36.	Lefebvre et al., 2020 [55]	CA	Other	T	Pre–post	Quality improvement	Speaking-up, conflict management, SCORE	Quant. survey	IIIb	Comm		++
37.	Letchworth et al., 2017 [56]	GB	Other	T	RT	T training, simulation	With T and M as per GAOTP + direct and closed-loop	Observer rating	IIb	Comm		++
38.	Lindhardt, 2014 [57]	DK	*Comm*	T	Pre–post	Comm training	Motivational interviewing, support behavior change	Observer rating	IIIb	Comm		+
39.	Lori et al., 2016 [58]	GH	*Comm*	M	Pre–post	Comm training	Health literacy-aware Comm	Mixed method	IIIa	Comm		+
40.	Lupi et al., 2012 [59]	US	*Comm*	S	RCT	Comm training, simulation	General + specific (e.g., completeness of info)	Survey + Observer rating	Ib	Comm		+
41.	Mancuso et al., 2016 [60]	US	*Comm*	T	Pre–post	Comm training	Quantity + quality (e.g., closed-loop Comm)	Observer rating	IIb	Comm		+
42.	Marzano et al., 2016 [61]	US	Other	T	Retrospective	T training, simulation	Standardization of comm; satisfaction with comm	Quant. survey	IIIb	Comm		+
43.	McArdle et al., 2018 [16]	US	Other	T	Pre–post	T training (TStepps) and clinical skills	SBAR, huddles, callout, checkback, closed-loop-; compliance with strategies	Quant. survey	IIIb	Comm		+
44.	Michelet et al., 2019 [62]	FR	Other	M	RCT	T training, simulation	Outcome measure: verbal exchanges	Observer rating	IIb	Verbal exchanges of T members		+
45.	Moore et al., 2020 [63]	ET	Other	R	Pre–post 11 m	SAFE-OB	Assist each other, T work, better T spirit	Mixed methods	IIIb	Twork + Comm		+
46.	Morony et al., 2018 [64]	AU	*Comm*	N/telehealth staff	RCT	Comm training	Quality of info received (e.g., sufficiency, usefulness, support of N)	Quant. surv (staff/caller)	IIa	Self-perceived Comm		+
47.	O’ Rourke, 2018 [65]	US	*Comm*	T	Pre–post	Patient safety interventions	Quality of hand-offs and Comm with M	Quant. survey	IIIb	Comm		+
48.	Phipps et al., 2012 [66]	US	Other	T	Pre–post	T training, simulation	Safety-related Comm, adverse outcomes index	Quant. survey	IIIa	Comm	Adverse outcomes	+
49.	Posner, 2011 [67]	CA	*Comm*	R	Pre–post	Workshop on error disclosure	Patient-centered (non-) verbal Comm	Observer rating	IIIb	Comm		+
50.	Raney et al., 2019 [68]	IN	Other	N	Retrospective	Simulation (PRONTO)	Structured clinical discussions and speaking-up	Semistruct. int.	IV	Satisfaction with training		+
51.	Régo et al., 2011 [69]	AU	Other	T	Pre–post	T training based on CRM	General Comm skills; calling for help	Mixed method	IV	Assertiveness, help seeking	Comm -	+
52.	Reszel et al., 2019 [70]	CA	Other	T	Retrospective	Patient safety culture	Emergency Comm strategies (e.g., SBAR)	Semistruct. int.	IV	Comm		+
53.	Riley-Baker et al., 2020 [71]	US	*Comm*	S	Pre–post	ACE.V in three simulated environments	Caring for M overall well-being; Comm appropriately with physician T members	Checklist	IV	Comm		+
54.	Romijn et al., 2019 [72]	NL	Other	T	RCT	T training	Intervention: SBARR Outcome measure: AOI	AOI	Ib	-	-	-
55.	Ronsmans et al., 2001 [73]	ID	Other	M	Retrospective	Comm + clinical skills	Comm behaviors (collecting, distributing info)	Structured interviews	IIIa	Info transferal		+
56.	Roter et al., 2015 [74]	US	*Comm*	M with low literacy, R	RT	Comm trainings	Patient and physician Comm behaviors	Observer ratings	IIb	M: online > f2f; R: opposite	R: online < f2f	+
57.	Santos et al., 2015 [11]	US	Other	T	Pre–post	Safety interventions	Standardized emergency comm + error reporting	Organizational data	IIIa	Staff report errors	Occurrence errors	+
58.	Sawyer et al., 2014 [75]	US	Other	R	Pre–post	T training, simulation	Comm among T members, calling for help	Observer rating	IIIa	Comm		++
59.	Shea-Lewis et al., 2009 [12]	US	*Comm*	T	Pre–post	T training (CRM)	Intervention: SBAR; Outcome: Adverse events	Organizational data	IIIb		Adverse outcomes	++
60.	Siassakos et al., 2009 [76]	GB	*Comm*	T	Pre–post, control group	T training, simulation	Comm behavior (command, enquiry, response, interruption etc.)	Observer rating qualitative	IIIb	Comm		+
61.	Siassakos et al., 2010 [77]	GB	Other	S	RCT	T training, (comm + simulation) vs. lecture	Within outcome: quality of Comm	Observer rating	Ib	Simulation > lecture: Comm		++
62.	Sonesh et al. 2015 [78]	US	Other	T	Pre–post	T training (TStepps)	Comm clarity and accuracy	Quant. survey	IIb	-	-	-
63.	Staines et al., 2019 [79]	CH	Other	T	Pre–post	T training (TStepps)	Comm openness, feedback + Comm errors	Quant. survey	IIb	-	-	-
64.	Thomas et al., 2010 [80]	US	Other	S	RT	T training, simulation	Comm (e.g., sharing info, inquiry, assertion); SBAR	Observer rating	IIIb	Comm		+
65.	Truijens et al., 2015 [81]	NL	Other	T, M	Pre–post	CRM T training	Comm effectiveness (e.g., SBAR)	Quant. survey	IIb	Comm at pregnancy only		+
66.	Walker et al., 2014 [82]	MX	Other	T	Pre–post	Simulation (PRONTO	Thinking out loud and clear, direct Comm	Observer rating	IIIb	Comm		+
67.	Walton et al., 2015 [83]	GT	Other	T	CT	Simulation (PRONTO)	Patient-centered Comm; effective Comm within T	Observer rating	IIb	Comm		+
68.	Warland et al., 2014 [84]	AU	*Comm*	S	Pre–post	Assertiveness training	Assertiveness (i.e., speaking-up)	Quant. survey	IIIb	Assertiveness		+
69.	Weiner et al., 2016 [85]	US	Other	T	Pre–post	Emergency (PROMPT)	Satisfaction with physician interaction rated by N	Quant. survey	IIIb	Satisfaction with physician interact.		+
70.	White et al., 2016 [86]	CD/MG	Other	R, M	Pre–post	Safety training	Part of the training but not defined	Semistruct. int.	IV	Comm (prior to the intervention)		+
71.	Zech et al., 2017 [87]	DE	Other	T	Pre–post	T training, simulation	Lack of Comm; openness of Comm	Quant. survey	IIIb	No overall change in openness of Comm		-

^1^ Study focus: Comm = communication; ^2^ Study population: M = mothers/patients, HCP = healthcare professionals, N = nurses, R = residents/medical doctors, S = students, T = teams; ^3^ TT = team training, ^4^ within the intervention and/or as the outcome; ^5^ OLE = Oxford Level of Evidence: IIIa: evidence from nonexperimental studies/inferential statistics, IIIb: evidence from nonexperimental studies/descriptive statistics, IV: qualitative studies, cf. Oxford Centre for Evidence-Based Medicine: Levels of Evidence (March 2009)—Centre for Evidence-Based Medicine (CEBM), University of Oxford; ^6^ Aggregated effects: ++ = effects as hypothesized, + = some effects, - = no effect. Further abbreviations in the table: ANTS = Anesthetists Non-Technical Skills; AOI = Adverse Outcome Index; CRM = Crew Resource Management; CTS = Clinical Teamwork Scale; EmONC = emergency obstetric and neonatal care; GAOTP = Global Assessment of Obstetric Team Performance; HSOPS(C) = Hospital Survey on Patient Safety (Culture); OSCE = objective structured clinical examination; MAAS-R (revised Maastricht history-taking and advice checklist); NOTSS (nontechnical skill for surgeons); NTS = nontechnical skills; PRIOR = practices in the operating room; covers in(ter)dependence, concern for self and others, dominance, conflict avoidance, integrating; PRONTO = Programa de Rescate Obstétrico y Neonatal: Tratamiento Óptimo y Oportuno; PROMPT = PRactical Obstetric Multi-Professional Training; RT = randomized trial; RCT = randomized controlled trial; RMC = Respectful Maternity Care; SAFE-OB = SAFE Obstetric Anesthesia course = 3-day refresher to address essential obstetric anesthesia and the most common causes of maternal death; SCORE (Safe and Reliable Healthcare’s safety, communication, operational reliability, and engagement; SBAR(R) = Situation Background Assessment Recommendation Read-Back; TStepps = TeamStepps [88].

## Data Availability

The data presented in this study are available on request from the corresponding author. The data are not publicly available due to legal and privacy issues.

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
