# Peer review of "Effectiveness of Communication Interventions in Obstetrics—A Systematic Review"

_ijerph, 2021, doi:10.3390/ijerph18052616_

Round 1

Reviewer 1 Report

Lippke and colleagues in their report study the effectiveness of communication training in obstetrics. The authors used the approaches of systemic review approach that involves peer-reviewed publication from 2000-2020. The outcomes of the study are not surprising however, are important.

Strengths of the study.

  1. Introduction is well written sufficient to understand the background of the study. The study design is robust and the search criteria are nice. In methodology, part authors should have clarified whether the search was performed by one author or more than one (Crosschecking is important here).
  2. Results: From the present study, authors have derived extensive information however choose not to discuss.
  3. Design of the study is robust , that involves several databases. Extensive literature search was performed.
  4. The way information was processed is an good example.
  5. The conclusions are well supported by the results obtained.

Points to improve:

The discussion needs to be improved. In addition, the following questions may help authors to improve their studies

  1. What was the pattern of communication training from 2000-2020? Does it improve with time? It should be discussed.
  2. Which methodology seems to be working best (Pre-post, Pre only, and CT) should be discussed.
  3. The need to include german literature in such analysis is difficult to understand.

Minor changes:

  1. Figure 1 should be improved. It would be nice to in the form of the flow diagram.
  2. Table 1: Authors should include the country where the research was performed. It will give an idea of the importance of communication training in different parts of the world.
  3. Table 1: Fist two references seem to be from the same group. It may create bias in the outcome of the study. It should be indicated in the footnote.
  4. Table 1: The effect column could have divided into two. Up and down arrow makes, it complicated to understand.

The study is well within the scope of the journal and maybe accepted for publication after the above-mentioned changes.

Author Response

REVIEWER 1

Comments and Suggestions for Authors

Lippke and colleagues in their report study the effectiveness of communication training in obstetrics. The authors used the approaches of systemic review approach that involves peer-reviewed publication from 2000-2020. The outcomes of the study are not surprising however, are important.

Strengths of the study.

  1. Introduction is well written sufficient to understand the background of the study. The study design is robust and the search criteria are nice. In methodology, part authors should have clarified whether the search was performed by one author or more than one (Crosschecking is important here).
  2. Results: From the present study, authors have derived extensive information however choose not to discuss.
  3. Design of the study is robust , that involves several databases. Extensive literature search was performed.
  4. The way information was processed is an good example.
  5. The conclusions are well supported by the results obtained.

Points to improve:

The discussion needs to be improved. In addition, the following questions may help authors to improve their studies

REPLY BY AUTHORS: Thank you very much for the positive feedback and the option to improve the discussion.

What was the pattern of communication training from 2000-2020? Does it improve with time? It should be discussed.

REPLY BY AUTHORS: According to our results, there was no timing effect. However, a quantitative analysis could follow this up and test accordingly as soon as enough studies exist for such an aggregation. We added this point to the results section which now reads “These effects appeared on an explorative level unrelated to the time of publication of the study.“ (lines 338f), and to the discussion: “Nevertheless, more research in terms of dismantling studies is needed. The same holds true regarding the relationship between the time of publication of the study and the effectiveness of the training. It would be worthwhile determining whether trainings improve over time due to more aggregated evidence feeding into improved training developments. Clearly, evidence should be used for informing future trainings and their evaluations.“ (lines 416ff).

Which methodology seems to be working best (Pre-post, Pre only, and CT) should be discussed.

REPLY BY AUTHORS: We appreciate this comment but are very hesitant to conclude on this as in the

Method section “3.1. Study characteristics according to the PICOS scheme”

Result section “3.3 Study quality according to the Oxford Level of Evidence” and

Discussion section “4.2.3 Comparisons and analyses”

We describe the different methodologies of the original studies and how this relates to the found effects.

However, we appreciate the reviewers’ question and have accordingly revised our discussion synthesizing the findings in this regard. Accordingly, this sentence now reads “…we can conclude that most of the pre-post comparisons indicate that trainings were effective. However, only 12 studies employed an RCT design, while the majority of intervention studies used designs with lower evidence levels. In addition, many lack a description of respective significance of results. Furthermore, only a few interaction effects were reported.

Training outcomes were oftentimes only measured in a post-treatment time point of measurement, with only one study providing more long-term evidence [23]. These results indicate a need for more well-planned, high-quality interventions with a clear description of training topics, methods, and corresponding outcomes. What seems promising, however, was that study quality was related to the occurrence of positive effects overall publications. Out of the study designs with a higher level of evidence (Ia to IIb), zero reported no or negative effects, while studies of lower quality were more likely to demonstrate no positive effects (8 out of 10 in evidence level IV).“ (lines 473ff).

The need to include german literature in such analysis is difficult to understand.

REPLY BY AUTHORS: Thanks, we included our rational for that explicitly: “Studies in the German language were included as the study was funded by a research funding and decision-making body in Germany interested in evidence. No further languages were included as no sufficient professional langue proficiencies existed other than English and German in the research team.” (lines 164ff).

Minor changes:

Figure 1 should be improved. It would be nice to in the form of the flow diagram.

REPLY BY AUTHORS: Thank you very much. We totally agree and used the standard form for reporting the study flow (please see the revised Figure 1).

Table 1: Authors should include the country where the research was performed. It will give an idea of the importance of communication training in different parts of the world.

REPLY BY AUTHORS: Thanks, we included a column with the country and have added the regarding information (please see revised Table 1).

Table 1: Fist two references seem to be from the same group. It may create bias in the outcome of the study. It should be indicated in the footnote.

REPLY BY AUTHORS: Thank you, we checked again the two studies and can confirm that they do not stem from the same data. Thus, we did not integrate a footnote and hope the reviewer agrees.

Table 1: The effect column could have divided into two. Up and down arrow makes, it complicated to understand.

REPLY BY AUTHORS: Thanks, we agree and have performed the according change.

The study is well within the scope of the journal and maybe accepted for publication after the above-mentioned changes.

REPLY BY AUTHORS: Thank you very much, we appreciate this positive feedback.

Author Response

Reviewer 2

While a lot of work went in to this review, the piece is much too long, in my opinion.

REPLY BY AUTHORS: Thank you for this feedback. We shorted as much as possible but do see only little option for shortening the manuscript further as all information is important and some aspects were still missing, which we needed to add, as pointed out by you below and the other expert reviewers. However, we removed all instances, where a shorting was possible and removed the description of the interventions into the Appendix. Thus, in the result section we now write “In the Appendix, we describe representative study designs grouped into highly effective interventions (section appendix 2a), those with moderate effects (Appendix 2b), qualitative research design (Appendix 2c), and studies not finding any effects (Appendix 2d). These effects appeared on an explorative level unrelated to the time of publication of the study.” (line 334ff).

Also, what does it add to the body of knowledge? For example Le Louis et al, Hodnett et al, McEwan et al published systematic reviews on the topic recently—some even published in 2020. I have no issues with the methodology of the systematic review; the search strategy is appropriate.

REPLY BY AUTHORS: We explicitly mention the uniqueness of this study to previous ones with the sentence now: “…we focus on communication as a part of teamwork and consider teamwork training if it includes a specific communication aspect (including digital interventions and simulation training) [20]. As no systematic review or meta-analyses could be found on explicitly this aspect, we fill this gap.“ (lines 109ff)

We are sorry to have missed out to clearly communicate what this manuscript adds to the body of knowledge. Thus, we added the value to previous studies by including the following: “…as this was not done before and previous systematic reviews within obstetrics focused on the training effects on teamwork and team performance [19] and hybrid simulation [6] or on interrelations of communication with obstetrics i.e. pediatric outcomes [1, 7, 10]. The previous reviews aggregating communication training programs match our findings but they were performed in general in midwifery [89], nursing care [14], or student learning [24]. Thus, our systematic review expands the previous state of science and will be synthesized in more detail in the following.” (lines 382ff).

However, I am not convinced that the researchers are able to separate the apples from the oranges—are studies included in the review truly similar on important factors? For example, the authors state in line 390 “they targeted diverse target groups and different aspects of communication”, which raises questions in my mind as to the comparability of studies. See also line 549 (“ it must be borne in mind that study designs and characteristics strongly differed between publications).

REPLY BY AUTHORS: We are sorry for this confusion and improved our description of the research aim not aiming to lumping together different factors but rather exploring the field carefully. Thus, we added the sentence to the research aims: “While the aim is to synthesize the evidence from the original studies, different factors relating to communication or target groups will not be separated. The included target groups and different outcomes, as well as various study designs and characteristics can be strongly differed between publications and will therefore only aggregated by means of a systematic review and not a meta-analysis.” (lines 132ff).

Also, are the published studies included in the review “one-shot” initiatives?

REPLY BY AUTHORS: Thank you for asking this. We report this in the column “Methodology” in Table 1, and added a concrete statement on that, too. “Thus, also investigations with just one measurement point (“one-shot” initiatives) were included.” (lines 244ff)

Did any have follow up over time to see if the interventions to improve communication actually lasted?

REPLY BY AUTHORS: Thank you for asking this. We now added this explicitly and the sentence now reads “Outcome variables could be measured via a pre-post comparison (i.e. follow up over time)…“ (lines 239ff) and “Only 6 publications reported interaction effects (8.5%), with only one study reporting a long-term follow-up” (lines 366f).

Usually with systematic reviews a forest plot is presented. Why was this not done here?

REPLY BY AUTHORS: A forest plot is untypical for systematic reviews but rather appropriate for meta-analyses. As we did not aim to quantitatively aggregate our findings, we did not include a forest plot.

Minor questions:

Introduction in lines 31,34, 35, 43 and actually throughout the text: why are italics used?

REPLY BY AUTHORS: This was done to improve readability but we understand that this is confusing so we formatted the text now without using italics.

Line 58: the statement is too declarative. Rewrite with “can play” to clarify that other things can and do contribute to patient safety

REPLY BY AUTHORS: Thank you, we have modified the sentence accordingly and it now reads “…patients can play a large role in causing such incidents and events.” (line 58f).

Reviewer 3 Report

Interesting topic. Well done. 

Comments regarding the manuscript :

Strengths:

-The use of many sources of information, and papers to do the review, means that it was a deep search they made.

-The use of PICOS was good.

-The topic was an interesting since communication training sometimes is took for granted in the sense that the safety of the staff and patients rely on the understanding of the instructions or recommendation from the professional to the patients. And its imperative to have a clear understanding in order to follow procedures and be safe at home.

Weaknesses:

-English need minor revision.

-The dates seems too apart from each other usually revisions or clinical trials are from 5-10 years or less.

Author Response

Review 3

Interesting topic. Well done. 

Comments regarding the manuscript :

Strengths:

-The use of many sources of information, and papers to do the review, means that it was a deep search they made.

-The use of PICOS was good.

-The topic was an interesting since communication training sometimes is took for granted in the sense that the safety of the staff and patients rely on the understanding of the instructions or recommendation from the professional to the patients. And it is imperative to have a clear understanding in order to follow procedures and be safe at home.

REPLY BY AUTHORS: Thank you very much for this positive feedback.

Weaknesses:

-English need minor revision.

REPLY BY AUTHORS: Thank you, we asked another colleague to proofread the manuscript and to eliminate all English grammar and typos. Accordingly, we added him to the acknowledgements.

-The dates seems too apart from each other usually revisions or clinical trials are from 5-10 years or less.

REPLY BY AUTHORS: Thanks, this is a good point which we very much appreciate to include. Accordingly, we added to the method section the following sentence: “Studies from the time frame of the last 20 years were included to get a large overview on the trainings and evidence from the last two decades’ developments. Moreover, this should level out outliers related to timing effects such as policy changes in specific countries, economic crises of specific regions or global challenges such as the COVID-19 pandemic.” (lines 170ff).

Round 2

Reviewer 2 Report

i think that a obgyn journal would be more appropriate for this piece.

Author Response

Review

Editor

In my recommendation, I think the authors should clearly indicate the advantages against the previous reviews in discussion or conclusion.  I understood the authors showed the differences of study design in lines 109-112 and 382-389.  However, I am afraid that the novel outcomes in the current study and their significances are still unclear.

REPLY BY THE AUTHORS: Thank you for pointing this out and we have accordingly included more elaboration into the manuscript now. The regarded introduction and discussion sections now read:

“Therefore, here we focus on communication as a part of teamwork and consider teamwork training only if it includes a specific communication aspect (including digital interventions and simulation training) [20]. As no systematic review or meta-analyses could be explicitly found on this aspect, but only on simulation training in obstetrics [6], teamwork in general [10,19], communication training in nursing care [14] or midwifery [21], we fill this gap by investigating all disciplines working in obstetrics (not only midwifes or nurses) and including all communication training approaches (not only simulation or teamwork).” (lines 109-116).

and

“With this systematic review, we aimed to aggregate the current state of research on communication interventions in obstetrics up until the end of the year 2020. This was done by looking at different target groups as this had not been done before. Previous systematic reviews within obstetrics generally focused on the training effects on teamwork and team performance [19], while in obstetrics, only hybrid simulations but no other forms of communication training were addressed [6]. While there are many studies including systematic reviews and meta-analyses, most of them simply aggregated interrelations of communication [1,7,10]. The previous reviews aggregating communication training programs match our findings but they were performed only generally in midwifery [21], nursing care [14], or student learning [25] but without an interdisciplinary and intersectoral approach as with our study. Thus, our systematic review expands the previous state of science and will be synthesized in more detail in the following.” (lines 384-395).

Reviewer 2

i think that a obgyn journal would be more appropriate for this piece.

REPLY BY THE AUTHORS: We agree and have chosen therefor the IJERPH “Special Issue: Obstetrics-Gynecology and Women's Health”. However, we agree with the reviewer that the findings should be replicated in other areas and have accordingly added to the conclusions “…and replicated in the future, and to also test whether these effects can be replicated in other settings relating to health promotion and prevention.” (lines 585f).

Furthermore, we have extensively improved the English with a native speaker (and included him into the Acknowledgments) and eliminated typos. Please see the current form of the manuscript with all adaptions highlighted in green.